# An elucidatory model of oxygen's partial pressure inside substomatal cavities

Andrew S: Kowalski[1,2]

[1]Department of Applied Physics, University of Granada, Granada, 18071, Spain
[2]Andalusian Institute for Earth System Research (IISTA), Granada, 18071, Spain

*Correspondence to*: Andrew S: Kowalski (andyk@ugr.es)

**Abstract.** A parsimonious model based on Dalton's law reveals substomatal cavities to be dilute in oxygen ($O_2$), despite photosynthetic $O_2$ production. Transpiration elevates the partial pressure of water vapour but counteractively depresses those of dry air's components – proportionally including $O_2$ – preserving cavity pressurization that is negligible as regards air composition. Suppression of $O_2$ by humidification overwhelms photosynthetic enrichment, reducing the $O_2$ molar fraction inside cool/warm leaves by hundreds/thousands of ppm. This elucidates the mechanisms that realize $O_2$ transport: diffusion cannot account for up-gradient conveyance of $O_2$ from dilute cavities, through stomata to the more aerobic atmosphere. Rather, leaf $O_2$ emissions depend on non-diffusive transport via mass flow forced by cavity pressurization, which is not negligible in the context of dynamics. Non-diffusive $O_2$ expulsion overcomes massive inward $O_2$ diffusion to force net $O_2$ emission. At very high leaf temperatures, mass flow also influences transport of water vapour and carbon dioxide, physically decoupling their exchanges and reducing water-use efficiency, independent of stomatal regulation.

## 1 Introduction

The partial pressure of water vapour (*e*) inside sub-stomatal cavities is well known to be greatly elevated by transpiration, as reflected by the ambient vapour pressure deficit (VPD). However, both the total pressure (*p*) and partial pressures of dry air components such as oxygen ($p_{O_2}$) have been fixed as parameters independent of plant functioning (e.g., Farquhar and Wong,1984). This oversight neglects the implications of Dalton's law of partial pressures. Here, a very simple model is presented that accurately estimates $p_{O_2}$, offering insights into the mechanisms of stomatal gas transport, which, contrary to long-standing assumption (Moss and Rawlins, 1963), are not exclusively diffusive.

## 2 Physical Law and Theory

Dalton's law of partial pressures,

$$p = e + (p_{N_2} + p_{O_2} + p_{Ar}), \tag{1}$$

30  defines $p$ as the sum of $e$ with the partial pressure of dry air, within the parentheses, which in turn is the sum of the partial pressures of nitrogen ($N_2$), $O_2$ and argon (Ar), neglecting gases with mere trace contributions. Equation (1) can be expressed for both the substomatal cavity interior ($i$),

$$p_i = e_i + (p_{i,N_2} + p_{i,O_2} + p_{i,Ar}), \tag{2}$$

as well as for the ambient atmosphere ($a$) outside the leaf,

35  $$p_a = e_a + (p_{a,N_2} + p_{a,O_2} + p_{a,Ar}). \tag{3}$$

If $\Delta$ denotes a cavity surplus versus ambient, subtracting Eq. (3) from Eq. (2) yields

$$\Delta p = \Delta e + (\Delta p_{N_2} + \Delta p_{O_2} + \Delta p_{Ar}), \tag{4}$$

where $\Delta e$ quantifies cavity humidification and reflects the ambient VPD. In the context of Eq. (4) for substomatal cavities, water vapour's substantial surplus ($\Delta e > 0$) implies either cavity pressurization ($\Delta p > 0$), or depressed partial pressures of dry

40  air's components ($\Delta p_{N_2} + \Delta p_{O_2} + \Delta p_{Ar} < 0$), or a combination of both. Since cavity pressurization would drive mass flow out of the aperture, theoretical considerations from micro-scale fluid dynamics can establish an upper limit for $\Delta p$.

Despite the fact that stomata are not cylindrical, the Poiseuille equation derivation (Giancoli, 1984) can be used to show that $\Delta p$ negligibly counterbalances $\Delta e$ in Eq. (4). This is done below by exaggerating the parameters of cylindrical geometry to put

45  a bound on the $\Delta p$ required to force viscous flow. The axial velocity $v$ of a laminar flow through a cylinder of length $L$ and radius $R$ is given as

$$v = \frac{\Delta p}{4\eta L} R^2, \tag{5}$$

where $\eta$ is air's dynamic viscosity (18 µPa s). Solving for $\Delta p$ yields

$$\Delta p = \frac{4\eta L v}{R^2}. \tag{6}$$

50  Here, parameters are chosen so as to maximize the $\Delta p$ required to drive viscous flow:

- Stomatal dimensions are exaggerated based on Lawson et al. (1998):
    - Pore depth is overestimated as $L = 10\mu m$,
    - Stomatal aperture is underestimated using $R = 2$ µm (area ~ 13 µm$^2$);

- An air velocity of $v = 6$ mm s$^{-1}$ escaping the stomatal aperture (Kowalski, 2017) represents an upper bound in the sense that traditional plant physiological models assume all transport to be diffusive, with no relevant role played by mass flow, effectively assuming a null value of $v$.

Plugging these values into Eq. (6) results in $\Delta p = 0.0011$ kPa, indicating that a very slight pressure difference is required to drive viscous flow. Given this, in the context of Eq. (4) regarding air composition and with resolution sufficient to characterize the VPD (to +/- 0.01 kPa), we can neglect substomatal pressurization in Eq. (4), taking $\Delta p = 0$. This means that any increase in the cavity's $\Delta e$ forced by transpiration must be counterbalanced by a reduction in the partial pressure of dry air ($\Delta p_{N_2} + \Delta p_{O_2} + \Delta p_{Ar} < 0$).

## 3 The Model

With transpired water vapour supplanting substomatal dry air, the simplest model is proportional depression of the partial pressures of dry air's components. In light of the Ideal Gas Law this implies that, for every 1000 dry air molecules displaced by water vapour, we can expect $N_2 : O_2 : Ar$ proportions of 781 : 210 : 9. Therefore $O_2$'s partial pressure inside substomatal cavities is modelled succinctly by

$$-\Delta p_{O_2} = 0.210 \cdot \Delta e, \tag{7}$$

indicating $O_2$ depression (versus ambient) that is 21% of the vapour pressure surplus of the substomatal cavity, or about 21% of the environmental VPD.

## 4 Model Implications, Accuracy, and Relevance to Other Gases

Oxygen deficits prevail within substomatal cavities because photosynthetic enrichment ($\mu$mol m$^{-2}$ s$^{-1}$) of $O_2$ is vastly overwhelmed by $O_2$ dilution and displacement due to transpiration (mmol m$^{-2}$ s$^{-1}$). Oxygen represents a sizeable fraction (about one-fifth) of ambient air but a far smaller fraction of leaf gas emissions, which are nearly pure water vapour and so dilute $O_2$ to force hypoxic conditions inside substomatal cavities. The degree of $O_2$ depression depends strongly on the VPD, and therefore leaf temperature ($T$), as illustrated by representative examples of cool and warm leaves (Table 1). Notably, even the cool leaf has a significant $O_2$ pressure deficit of $-\Delta p_{O_2} = 0.066$ kPa. "Near sea level" (defined hereinafter as $p = 100$ kPa), this corresponds to an $O_2$ molar fraction (referencing moist air) that is 660 ppm below ambient. In warm leaves $O_2$ depression reaches several thousand ppm, and in torrid environments it can be far greater.

|  | **Cool** | **Warm** |
|---|---|---|
| $T_{\text{leaf}}$ | 10ºC | 34 ºC |
| $T_{\text{air}}$ | 8ºC | 30ºC |
| $e_{\text{leaf}}$ | 1.228 kPa | 5.325 kPa |
| $e_{\text{air}}$ | 0.912 kPa | 3.610 kPa |
| $\Delta e$ | 0.316 kPa | 1.715 kPa |
| $-\Delta p_{O_2}$ | 0.066 kPa | 0.359 kPa |
| $-\Delta \chi_{O_2}$ | 660 ppm | 3590 ppm |

**Table 1: Consequences of negligible stomatal-cavity pressurization regarding air composition. Representative temperatures, water vapour pressures, stomatal cavity vapour pressure surplus ($\Delta e$), oxygen pressure deficits ($-\Delta p_{O_2}$), and oxygen concentration deficits ($-\Delta \chi_{O_2}$) for cool and warm leaves and their ambient atmospheres. Leaves are taken as saturated and ambient air at 85% relative humidity; $-\Delta p_{O_2}$ is calculated using Eq. (7); $-\Delta \chi_{O_2}$ is calculated for conditions "near sea level" ($p = 100$ kPa).**

The most noteworthy inference from this Daltonian model regards the mechanisms of gas transport through stomata, since $O_2$ produced by photosynthesis cannot diffuse out of stomata as has been traditionally assumed (Parkhurst, 1994). Equation (7) implies that substomatal cavities are generally much more dilute in $O_2$ than their environments, whatever the leaf $T$. Although traditional thinking would explain $O_2$ transport in terms of diffusive flows within a ternary system (Jarman, 1974; von Caemmerer and Farquhar, 1981), diffusive transport from dilute towards enriched regions is impossible – it would violate the $2^{\text{nd}}$ Law of Thermodynamics. Rather, non-diffusive transport by the viscous flow – driven by pressurization that is negligible in the context of Eq. (4) but nonzero nonetheless – is required to overcome inward $O_2$ diffusion and drive $O_2$ out of substomatal cavities. Diffusion of $O_2$ into substomatal cavities is massive, due to concentration differences of hundreds or thousands of ppm across the leaf's pore depth. Gradients and diffusion of $O_2$ exceed those of $CO_2$ by orders of magnitude.

However simplistic, the model improves upon the accuracy of previous assumptions regarding substomatal $p_{O_2}$ that neglected Dalton's law. These include the assumption that $p_{O_2}$ is a fixed parameter that does not depend substantially on plant functioning (Farquhar and Wong, 1984), as well as the notion that substomatal cavities are enriched in $O_2$ (Parkhurst, 1994), purporting outward $O_2$ diffusion while overlooking the dominant effects of transpiration on $O_2$ abundance. The greatest inaccuracies of the Daltonian model presented here can be bounded by considering the chief processes that it does not take into account.

Adhering to the principle of parsimony, the model neglects the effects of two lesser and partially offsetting influences on $p_{O_2}$, neither of which can alter the above conclusion regarding $O_2$ transport mechanisms. Firstly, photosynthetic $O_2$ production must reduce the $O_2$ pressure deficit, increasing substomatal $O_2$ somewhat, but certainly not by the many hundreds of ppm (or

thousands for warm leaves) that would be required to make $\Delta p_{O_2}$ positive. This seems clear when recalling the stoichiometric relation between $O_2$ and $CO_2$, and the trace amounts of the latter gas that limit the possible magnitude of photosynthetic $\Delta p_{CO_2}$. Secondly, molecular diffusion's discrimination among dry-air species must increase the $O_2$ deficit since $N_2$ (28 g mol$^{-1}$), representing 78.1% of atmospheric dry air molecules, diffuses upstream into substomatal cavities more rapidly than does $O_2$ (32 g mol$^{-1}$) according to Graham's law. Unaffected by these inaccuracies, the deduction that substomatal cavities generally are very dilute in $O_2$ is ineluctable, as is the conclusion that stomatal $O_2$ transport is predominantly non-diffusive. Specifically, it is due to a mass flow that indiscriminately pushes all gases outwards (Kowalski, 2017). Although previously couched in terms of "stomatal jets", this is a low-velocity, viscous flow (low Reynolds number) whose conveyance neither discriminates among gas species nor depends on concentration gradients, unlike diffusion. Its relevance to the transport of other gases depends on air's state conditions within stomata.

At very high leaf $T$, these implications from gas physics become relevant to the behaviour of $CO_2$ and water vapour.

Regarding $CO_2$, non-diffusive transport cannot be neglected universally. The $p_{O_2}$ model presented here is not valid for estimating $p_{CO_2}$, whose fluctuations are principally determined by photosynthesis. However, independent of photosynthetic drawdown (well, physically independent), the assumption of proportional depression of the partial pressures of dry air's components when supplanted by water vapour seems valid. Accordingly, just as Eq. (7) apportions 21% of supplanted dry air to $O_2$ depression, for a $CO_2$ concentration of 420 ppm we can expect 0.042% of the dry-air depression described by Eq. (4) to correspond to $p_{CO_2}$. This influence is negligible for temperate leaves with modest VPDs. For example, for the cool leaf in Table 1, it implies $CO_2$ depression of ~0.0001 kPa; near sea level, this is about 1 ppm and pales in comparison to photosynthetic drawdown. By contrast, for the warm leaf also near sea level, it means substomatal $CO_2$ depression by over 7 ppm, which is no longer negligible and drives inward $CO_2$ diffusion that is not due to photosynthesis. During heat waves, with extreme values of VPD, substomatal $CO_2$ depression due to humidification can be much larger. Thus, at very high leaf $T$ non-diffusive transport can appreciably suppress photosynthesis via $CO_2$ limitation, but it has the opposite effect on transpiration.

Water vapour is also forced out of stomata by non-discriminating mass flow, with relevance that depends on water vapour abundance. Applying Newtonian physics to the momentum of air within stomata, Kowalski (2017) showed that the water vapour mass fraction, or specific humidity ($q$), defines the fraction of water vapour transport that is non-diffusive. Within substomatal cavities that are essentially saturated, the state variable $q$ is largely determined by $T$. For the cool leaf in Table 1 ($q < 1\%$), non-diffusive transport can reasonably be neglected. But this is not so for the warm leaf ($q > 3\%$), and furthermore $q$ increases rapidly as leaf $T$ rises. If these increases in water vapour transport rates seem modest, versus what can be achieved by diffusion alone, they grow in importance when considered in combination with reduced photosynthesis via suppression of sub-stomatal $p_{CO_2}$ by $\Delta e$ as described above.

The consequences of gas physics at high leaf $T$ are disparate for water vapour and $CO_2$ exchanges. Ejecting all gases, mass flow enhances water-vapour loss and opposes $CO_2$ ingress, boosting transpiration and suppressing photosynthesis versus the capabilities of diffusive transport alone. It thereby reduces water-use efficiency via effects on each gas. Therefore, dry-air depression and non-diffusive transport likely explain the decoupling of transpiration and photosynthesis that has been observed widely at very high leaf $T$ (Aparecido et al., 2020; De Kauwe et al., 2019; Diao et al., 2024; Krich et al 2022; Marchin et al., 2023; Pankasem et al., 2024; Sun et al., 2024). In very hot substomatal cavities where water vapour is not a mere trace gas, transport due to mass flow casts doubt on the very meaning of stomatal conductance. And non-diffusive transport is gaining in relevance regarding leaf gas exchanges as the Earth warms and heatwaves increase in frequency and intensity (IPCC, 2021).

## 5 Prospects for Unveiling Stomatal Fluid Mechanics

Evaporation within a moist cavity and vapour egress through a small aperture aptly describes not only leaf gas exchanges but also a whistling tea kettle. At the boiling point, a steam jet drives out dry air (including $O_2$ and $CO_2$) and the water vapour pressure approaches the total pressure ($e \sim p$). This marks the humid extreme ($q \sim 100\%$) of a spectrum regarding fractional transport by different mechanisms, with the diffusion-only modelling framework valid at the other extreme (dry; $q \sim 0\%$). In state conditions that categorise stomatal air, $q$ is limited to below 10% and non-diffusive transport plays a role that is secondary, although sometimes not negligible. Insight into the consequences of such mass flow might be gained by investigating gas exchanges at intermediate values of $q$.

Artificial experiments may be helpful in this regard and there are several strategies that can elevate $q$, and that can be pursued individually or in combination. Gas exchange measurements can be made at very high temperatures (exceeding 50ºC) using artificial leaves (Schymanski and Or, 2017) since they suffer no heat stress or loss of functionality under conditions that would endanger life. Evaporation from such leaves with rising $q$ but at constant VPD is predicted to be practically constant according to stomatal conductance models, but to increase when taking non-diffusive transport into account. Similar experiments might be conducted on living leaves with hot but tolerable temperatures in conditions nearer to boiling due to reduced $p$, as within a hypobaric chamber. Finally, for leaves functioning in a "helox" environment (Mott and Parkhurst, 1991) – a mixture of helium and $O_2$ whose density (hence inertia) is just 29% that of dry air – non-diffusive transport would be elevated more than threefold (Kowalski, 2017). Assessments of leaf functioning in such conditions should help to shed light on the implications of non-diffusive transport to stomatal gas exchanges.

## 6 Conclusions

Water vapour's elevated partial pressure inside substomatal cavities implies depressed partial pressures of dry air components including oxygen ($O_2$), according to Dalton's law with negligible cavity pressurization. Substomatal cavities, not photosynthetically enriched in $O_2$, are dilute because of transpiration. Only non-diffusive conveyance can account for transport of $O_2$ from these $O_2$-poor cavities into the more aerobic, ambient atmosphere. Slight substomatal pressurization, however negligible in the context of Dalton's law, is sufficient to drive mass flow of air out of stomatal apertures. The relevance of mass flow to gas transport cannot be neglected universally in plant physiology, becomes important for water vapour and $CO_2$ in leaves at very high $T$, and therefore is increasing with global warming.

## Competing interests

The author declares that he has no competing interests.

## Acknowledgements

The author is supported by Spanish government projects PID2021-128463OB-I00 (REMEDIO), Ref: 2822/2021 (EVIDENCE), PID2020-117825GB-C21 (INTEGRATYON3), PN2021-2820s (IBERALP), and TED2021-129499A-I00 (MANAGE4FUTURE), as well as University of Granada projects PPJIB2022-08 (MODELICO) and C-EXP-366-UGR23 (MORADO) including European Union ERDF funds.

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
