# Peer review of "An elucidatory model of oxygen's partial pressure inside substomatal cavities"

_EGUsphere, 2024_

## Author Response (AR1)

Please note that this "author's response" is simply a copy/paste of my replies to the reviewers in the on-line discussion. The changes made as a consequence are those proposed in my replies to Reviewer 1.

**Reviewer 1**

I thank the anonymous referee for this constructive assessment of the manuscript, since several of the comments have enabled me to propose revisions that should improve clarity. The comments are repeated below in italicized font, followed by my replies in normal font.

*This manuscript presents a theoretical, physically based model to explain the possible occurrence of nondiffusive exchange of gases between plant leaves and the atmosphere. The approach is founded in fluid dynamics and makes some simple assumptions in arguing that plant physiologists have been incorrect in their treatment of gas exchange processes. The article is really more like an opinion piece in the way it's written, starting from the first sentence of the introduction. If the author wants to develop a dialog, a more moderate tone might help.*

For decades now, reviewers have admonished me regarding the "tone" of my writing. Clearly it is a defect, and one that I have worked to improve, but apparently without great success. I am very much open to any specific suggestions that would improve the tone without changing the scientific meaning.

*The paper could be improved by starting with a clear statement of the evidence that there is a potential mistake in the conceptual framework underpinning leaf gas exchange, specifically the assumption that stomatal fluxes are always diffusive. The broad context of the "decoupling" of photosynthesis and transpiration is a widely observed phenomenon especially in response to heat waves. The author has developed a possible physical explanation for this observation. The paper could also be improved by clarifying a number of issues as described below.*

The introduction section, a lone paragraph of just four sentences, is intented as "a clear statement of the evidence that there is a potential mistake in the conceptual framework underpinning leaf gas exchange, specifically the assumption that stomatal fluxes are always diffusive." I have no doubt that other authors could state it more clearly, but have done my best in this regard.

*Specific comments*

*L9. "preserving cavity pressurization that is negligible as regards air composition." This part of the sentence could be a separate sentence because the total cavity pressure is an important and separate consideration from the partial pressures. Indeed, cavity pressurization caused by transpiration appears to be the main mechanism for oxygen emissions, according to the theory.*

Total cavity pressure is not separate consideration from the partial pressures, but rather directly linked via Dalton's law. This is the very basis of my analysis as reflected by the fact that it is mentioned in the first sentence of the manuscript.

*L10. Change "suppression" to "dilution"*

There are two processes that lead to depression of oxygen. Dilution is certainly one, but the other is displacement. Perhaps an examination of a stomatal cavity in light of the Ideal Gas Law will clarify this.

Consider a cavity of constant volume, and for simplicity at constant temperature, with an initial pressure ($p1$) and containing $n1$ moles of air. If transpiration adds a single water vapour molecule

to the cavity, then some other molecule (very probably nitrogen) must exit, since n2>n1 implies pressurization (p2>p1) via the Ideal Gas Law, driving outward flow until pressurization is exhausted. Ultimately, the cavity is both enriched in water vapour and depleted in nitrogen, which will then tend to diffuse back inward while water vapour diffuses outward.

Now if we repeat this experiment five times, statistically we can expect expulsion of four nitrogen and one oxygen molecules. Oxygen is both diluted and displaced by transpiration. Note that in steady-state conditions, nitrogen is displaced outward but diffuses inward, and the two transport mechanisms cancel out such that there is no net transport of nitrogen (except, perhaps, for legumes or other nitrogen-fixing plants; I am no expert on nitrogen uptake). The same argument could be made regarding oxygen, but then we must take into account photosynthetic enrichment.

If we repeat the experiment a million times, then things get interesting for the vital gases. Of the million displaced molecules:

- about 780,000 will be nitrogen;

- nearly 210,000 will be oxygen, far exceeding what photosynthesis can add;

- thousands will be water vapour, whose departures represent non-diffusive vapor transport because they were pushed out;

- hundreds will be carbon dioxide, whose depletion (and subsequent diffusion) are not due to photosynthesis.

To me, it is clear that dilution is not the only process that reduces substomatal oxygen. Displacement also does so, implying non-diffusive transport. This is the point of the manuscript, and therefore I prefer not to change "suppression" to "dilution".

*L 59, It is confusing to say that cavity pressurization can be neglected, because in the abstract it is implied that cavity pressurization is not negligible.*

Context is very important. Clearly the same phenomenon can be negligible in one regard but not in another. For example, at 2 ppm methane plays a negligible role in determining the specific heat of air, but not in absorbing infrared ratiation.

The abstract says that cavity pressurization is "negligible **as regards air composition**" (line 11), but "not negligible **in the context of driving viscous flow**" (line 18; bold emphasis added in each case). To reduce confusion, I propose to modify the sentence mentioned at L 59 begins to specifically state that the context in which pressurization can be neglected is that of air composition,  because it is about to be used in Eq. (7) to model the partial pressure of oxygen.

*L 90, here we see that pressurization is negligible but non nonzero…*

This is because the context of Table 1 is that of air composition, not dynamics. Again, to make this more clear, I propose to change the Table caption so that it begins with "Consequences of negligible stomatal-cavity pressurization regarding air composition."

*L 108110, I am not convinced that the substomatal cavity is dilute in oxygen. Inward diffusion will offset the vapor effects. It would be nice to see some evidence for this theory, such as in the isotopic composition of oxygen or carbon dioxide that might be altered in the process of biosphereatmosphere exchange.*

Substomatal cavities are dilute in oxygen because leaf gas emissions are dilute in oxygen. As I have written elsewhere in Copernicus open discussions (https://doi.org/10.5194/bg-2023-30-CC1), even with a modest evaporation rate and robust photosynthesis, less than 2% of the molecules emitted within substomatal cavities are oxygen (versus 21% in the atmosphere). (More than 98% are water vapour.) Thus, the combination of transpiration and photosynthesis results in dilution of stomatal cavities, driving inward oxygen diffusion.

Evidence of this in the isotopic composition of oxygen has long existed and is known as the "Dole effect", with tropospheric air richer in heavy oxygen than seawater (Dole, 1935). The massive inward diffusion of oxygen results in fractionation, with only the lightest oxygen molecules able to diffuse upstream against the jet and so be consumed by respiration within the stomatal cavity, leaving the atmopshere enriched in heavy oxygen. I did not include this in the paper because it seems likely to cause even more controversy.

Dole, M., 1935, The relative atomic weight of oxygen in water and in air, J. Chem. Phys., 4(4), 268-275.

*L 134135, It is hard to understand this sentence "If these increases in water vapour transport rates seem modest, versus what can be achieved by diffusion alone, they grow in importance when considered in combination with jet suppression 135 of photosynthesis."? Until one reads L 137139. This is another example of how the paper could be improved by putting the problem statement before the solution, instead of viceversa.*

I generally appreciate suggestions to improve the structure of my writing, but in this case cannot understand the referee's meaning.

This section of the paper addresses the influence of jets on transport of carbon dioxide and water vapour. It has three paragraphs whose themes appear in the first sentence of each: the first (L 116) addresses carbon dioxide, the second water vapour (L 129), and the third their relative behaviour (L 137) including water-use efficiency and decoupling. The sentence cited by the referee, as the last in the paragraph about water vapour, is intended guide the flow of the text to the following paragraph.

Perhaps if the referee could clarify what is meant by "problem statement" and "solution", I could understand how to improve this section of the paper.

*L 144145, It seems unnecessary to "cast doubt on the very meaning of stomatal conductance." Certainly the concept is useful, as is the concept of hydraulic conductance as it pertains to water flow through porous media, where both diffusion and mass flow occur.*

I disagree. Stomatal conductance describes diffusive fluxes as a function of concentration gradients, while hydraulic conductance describes non-diffusive fluxes as a function of pressurization. Physically, they are very dissimilar.

Physiologists often interpret stomatal conductance as meaning the degree to which stomata are open (e.g., Urban et al., 2017). Strictly speaking, however, it is a ratio of flux to concentration difference. In the diffusion-only paradigm, stomatal conductance for water vapour and carbon dioxide are coupled (via Graham's law), and covary with the degree of stomatal aperture. But non-diffusive transport, when not negligible, makes this not so. Within a stomatal aperture of fixed dimensions, a temperature increase to extreme values invalidates the paradigm, enhancing water vapour egress but inhibiting carbon dioxide ingress. This is best illustrated by the case of boiling (where the vapour pressure equals the total pressure, yielding a specific humidity of 100%). Gas exchange through the spout of a boiling tea kettle has nothing to do with diffusion:

- water vapour is the only gas present, and so cannot diffuse with no concentration gradient but has a large flux (infinite stomatal conductance) that is non-diffusive in nature (i.e., a jet);

- carbon dioxide cannot enter the kettle despite an enormous concentration gradient (zero stomatal conductance) because the jet is so strong that it cannot diffuse upstream.

(Recall that boiling can be achieved, either by raising the temperature at constant pressure, or by lowering the pressure at constant temperature.)

When state conditions tend towards boiling, stomatal conductance for water vapour increases but this does not mean that the aperture has expanded, and stomatal conductance for carbon dioxide decreases but this does not imply that the aperture has contracted. I believe, therefore, that jet transport casts doubt on the meaning of stomatal conductance.

Urban, J., et al., 2017, Increase in leaf temperature opens stomata and decouples net photosynthesis from stomatal conductance in pinus taeda and populus deltoides nigra, J. Exp. Bot., 68, 1757-1767.

**Citation**: https://doi.org/10.5194/egusphere-2024-1966-AC1

**Reviewer 2**

I thank the anonymous referee for the engaging discussion. Although pushing back against suggested changes, I appreciate this on-line forum as a space in which to rationalize decisions regarding wording and composition, and remain open to further suggestions to improve the manuscript. The referee's comments (italic font) are followed by my replies (normal font).

*I agree that the model of stomatal flux as exclusively diffusive is oversimplified; at times I've said that one person's diffusion is another's advection, and applying advective flux principles to stomata is likely to lead to new insights.*

I appreciate the reviewer's perceptive perspective.

*The parentheses aren't needed in equations 1-3, and 'mere trace' should perhaps be quantified more explicitly, perhaps as greater than 0.9% (argon) or 0.4% (water, on average), because many peoples' research careers depend on these key trace gases including CO2!*

Mathematically, the parentheses do not change these equations, and so they are not needed. However, they are included to focus attention on the partial pressure of dry air (line 30). I find them helpful as a reminder that, with negligible cavity pressurisation, humidification suppresses the partial pressure of dry-air and thereby its every component.

I worry that an attempt to quantify "mere trace" would open a can of worms. The most abundant trace gas is $CO_2$ at about 0.042% of the atmosphere. Its neglect would mean describing atmospheric composition with 99.958% accuracy, which seems acceptable. But where should we delineate the acceptable degree of accuracy? Can we confidently say that 99% is acceptable, but 97% is not? I have searched the literature, of not only atmospheric gases but science in general, for some definitive and citable statement regarding what is and is not negligible. I have found nothing concrete, and so hesitate to explicitly quantify "mere trace".

This sentence begins with the words "Dalton's law of partial pressures", which establish its context. As I have argued to the other reviewer, context is very important when determining what can and cannot be neglected. In the context of Dalton's law, $CO_2$ is negligible. Of course, it is not negligible in the context of atmospheric absorption of infrared radiation, but that is another story altogether.

*In equation 4, the delta implies a surplus or a deficit depending on direction (the text only states surplus)*

Line 36 defines Δ as denoting a surplus. Mathematically, a negative surplus is a deficit. Thus, $\Delta e > 0$ describes a surplus of water vapour, whereas for dry-air component $i$, $\Delta p_i < 0$ describes a deficit.

*54: This value still seems quite fast, even as an upper bound, but could be justified in more detail by explaining a bit more the contents of Kowalski 2017 as it applies to the present study*

An earlier draft of this manuscript included the following text:

> Newtonian physics identifies an air jet escaping stomata (Kowalski, 2017), here summarised. Air components have diverse momenta (kg m s$^{-1}$) including outward $H_2O$ and $O_2$, null $N_2$ and Ar, and inward $CO_2$. Air's momentum is the sum of its components' momenta. With $H_2O$ dominating stomatal gas exchange by orders of magnitude, air's momentum density equals the $H_2O$ flux density ($F_{H_2O}$; kg m$^{-2}$ s$^{-1}$) that quantifies transpiration. The outward airspeed is therefore the momentum-to-mass ratio $\frac{F_{H_2O}}{\rho}$

A colleague who helpfully read the manuscript suggested that, for a paper describing a model of oxygen's partial pressure inside stomata, this text is extraneous and distracting.

The purpose of this section of the paper is to establish that cavity pressurisation is negligible in the context of Dalton's law, **at most** $\Delta p = 0.0011$ kPa. The reviewer says that "*This value still seems quite fast*", implying that I have likely overestimated $v$ and therefore $\Delta p$. If anything, this buttresses the argument that pressurisation can be neglected ($\Delta p = 0$). Therefore, I think my colleague was correct, and that there is no need to justify in more detail using $v = 6$ mm s$^{-1}$ as an upper bound.

*I'm having trouble fully following the paragraph on line 57. The difference is about 1 Pa, which isn't much. Is the argument that, even with extreme parameter values there is little reason to believe that delta_p is approximately zero such that positive delta_e implies negative delta_p of the other gases?*

I think the reviewer follows perfectly, but also that example values might illustrate this more clearly. Let's consider a "moderate VPD" (Aliniaeifard and van Meeteren, 2014), with $\Delta e = 1.17$ kPa, and examine two cases when assessing dry air's partial pressure, between parentheses in Eq. (4):

1. If we use the upper bound of $\Delta p = 0.0011$ kPa, then it must be depressed by 1.1689 kPa;
2. If we neglect pressurisation and use $\Delta p = 0$, then it must be depressed by 1.17 kPa.

These outcomes are 99.9% the same, which I think justifies assuming $\Delta p = 0$ in Eq. (4). I do not wish to add these example values to the manuscript, unless the reviewer and editor so prefer.

**Reference**

Sasan Aliniaeifard, Uulke van Meeteren, Natural variation in stomatal response to closing stimuli among *Arabidopsis thaliana* accessions after exposure to low VPD as a tool to recognize the

mechanism of disturbed stomatal functioning, *Journal of Experimental Botany*, Volume 65, Issue 22, December 2014, Pages 6529–6542, https://doi.org/10.1093/jxb/eru370

*I still find the evidence to be by omission: if O2 can't leave stomatal cavities by diffusion, it must be 'jets'. But what do these jets look like, how do they 'burst' and do other mechanisms like Bernoulli pumping cause the transport in practice? To me the major weakness is a lack of description of how the jets work, for lack of a better word, in practice and if fluid mechanical simulations, conceptual models, or studies of 'bursting' when stomata open (or the 'thermostat' model that Joe Berry described) would be most fruitful for better understanding stomatal dynamics going forward.*

Rejecting long-embraced ideas can be a major challenge for advancing knowledge (Keynes, 1936), and I believe this applies to the assumption that diffusion alone transports gases through stomata. I would argue that, if $O_2$ cannot leave stomatal cavities by diffusion, then non-diffusive transport must be responsible. The terminology of "jets" may be appropriate as argued by Kowalski (2017), or it may not. Once it is accepted that non-diffusive transport exists, science can then address its mechanistic functioning as a next step. Does it occur as a steady stream, or in spurts? If the latter, are the spurts triggered by atmospheric turbulence (a Bernoulli effect), or by thermodynamic forcings, or what? Such questions certainly seem relevant to understanding stomatal control of gas exchanges, particularly at high temperatures.  I share the referee's curiosity regarding these questions, but feel that they will be best addressed only once the scientific community accepts the relevance of non-diffusive transport.

**Reference**

Keynes, J. M., The General Theory of Employment, Interest and Money, Macmillan (1936)

---

## Author Response (AR3)

**15 Oct 2024: Anonymous Referee #2, Report #1**

*Kowalski describes a model for oxygen transport from stomata that notes that Dalton's law implies that oxygen concentration within the stomatal cavity must be below ambient. The work I interesting but there are still a few points that need to be clarified and reconsidered.*

I thank the anonymous referee for evaluating the manuscript.

*A schematic could be helpful but perhaps even more so is the acknowledgement that plants also consume oxygen if their cells have mitochondria; oxidative respiration to drive cellular function has to come from somewhere. Obviously the respiratory needs of plants are rather small compared to most animals but worth mentioning for completeness as the results have implications for plant cellular function.*

The referee suggests acknowledging that plant cells consume oxygen. I am afraid that this misses the point of the manuscript, which regards gaseous air spaces and not cell tissues. During active leaf functioning, the photosynthesising cells that line stomatal cavities must be well oxygenated, beyond their respiratory requirements, because they emit oxygen. However, their water vapour emissions exceed those of oxygen by orders of magnitude. Oxygen represents a sizeable fraction (about one-fifth) of ambient air but a far smaller fraction of leaf gas emissions, which are nearly pure water vapour and so dilute $O_2$ to force hypoxic conditions inside substomatal cavities. I have added this explicatory, underlined sentence to the beginning of Section 4, and hope that it helps the referee and future readers to focus on gases within substomatal cavities.   Any discussion of cellular oxygen would distract and confuse the issue, and is best avoided.

*On 38: delta_e here may not reflect the ambient VPD as commonly assumed if recent studies by Cernusak and colleagues regarding the non-saturation of the sub-stomatal space hold true. (There certainly is a surplus as noted on line 39).*

[Figure]

The referee suggests that $\Delta e$ might not reflect the VPD, if the sub-stomatal cavity is not completely saturated. I am afraid I disagree. Here, the verb "equal" was discarded in favour of "reflect", which has a less precise meaning. For example, in the image at right, the lake reflects the mountains and sky. It does so imperfectly since rocks are visible at the lake's bottom (it is not a mirror), but it does reflect. I think that the word "reflect" is correctly chosen here. This is also why the last sentence of section 3 (The Model) says "… 21% of the vapour pressure surplus of the substomatal cavity, or **about** 21% of the environmental VPD" (emphasis added here), allowing for possible under-saturation.

*On 111 and elsewhere regarding the air jet, given that previous work is alluded to here rather than described is this related to the Bernoulli principle?*

The referee asks whether the airflow is related to the Bernoulli principle. I do not believe that this is a relevant framework. The Bernoulli principle derives from the assumption of an inviscid fluid. Neglecting the effects of friction, it expresses energy conservation as a conversion between three types of energy: pressure-volume work, gravitational potential energy, and kinetic energy.

The tiny stomatal size and low velocity give the airflow exiting stomata a low Reynolds number. In such viscous flows, conversion of kinetic energy into molecular kinetic energy (by dissipation, raising the temperature) cannot be neglected in the equation of motion. This invalidates the Bernoulli derivation; in this viscous case, the Bernoulli principle is not an appropriate framework for estimating velocities from pressure differences.

However, I thank the reviewer for this comment because it has opened my eyes to the misleading effects of the terminology of "stomatal jets". Although I published previously using this terminology (Kowalski, 2017), I have come to realise that it is unhelpful. The word "jet" has high-velocity connotations and may be misleading in this context. Air does not *jet* out of stomata; it *oozes* or *seeps* (which seem like better verbs for describing viscous motion). I have therefore revised the manuscript to correct this unhelpful terminology, and substituted "mass flow" for "jet" where appropriate.

**5 Dec 2024: Anonymous Referee #3, Report #2**

This report contains no comments that I can find, and a recommendation that the manuscript be accepted as is. Having served as editor for a different journal, I appreciate that such a review is of little help, neither to the editor nor to the author in revising for improvement. At least, however, it is an unambiguous opinion.

**12 Dec 2024: Associate editor decision: Publish subject to minor revisions (review by editor)**

*I am now in receipt of two additional reviews who are generally favorable about the manuscript and its scope but discuss minor revisions that I feel will help clarify key points. Please consider these comments and please don't hesitate to reach out if questions arise. I am posting Reviewer #2's comments below as they may not be visible by the author in the Copernicus system as uploaded.*

Again, I appreciate the editor's efforts in handling this manuscript. My replies to Reviewer #2 appear on previous pages at the beginning of this document. Since the comments below seem to come from a different scientist (they refer to reviewer 2), I will refer to them as having come from Reviewer X.

*I read the revised manuscript by Kowalski and their response to reviewers. Kowalski addresses their scientific concerns. I think they could engage more with the reviewers' comments about tone and understanding, as the reviewers' comments likely reflect the broader sentiment of potential readers. However, I tend to defer to the author to let them communicate how they feel is best. I do not have a a strong background a fundamental physics, but instead have learned along the way as I became interested in plant ecophysiology. Hence, I found the manuscript interesting, very clear, and I learned from it. My gut reaction is that if non-diffusive transport was important, wouldn't there be a large body of observations that were inconsistent with existing models? I recognize this isn't very sound logic on my part, but the paper would be more convincing if this could be demonstrated. Alternatively, a quantitative demonstration that jets can explain the decoupling between A and E at high temperatures would be more convincing. This decoupling is an interesting observation, but seems like it could be readily explained by other factors such as deactivation of enzymes either by regulation or denaturation, for example. I also wish that Kowalski engaged with reviewer 2's comment about a clearer physical description of what is going on with these jets. To extend that, I would like to know what sort of experiments or measurements should be done to work out the significance of O2 jets. It's not clear to me if the physics is so well validated that we can trust the model or if there are hidden assumptions or oversimplifications that are being neglected. I do not necessarily think all of these points need to be addressed in the revision, but as a practicing ecophysiologist, addressing these points would make the argument more convincing.*

To clarify the subjects of my replies, I have highlighted five sections above of the comments from Reviewer X:

1. Tone and understanding: I have consulted with a highly regarded colleague, and received and heeded some advice on how to improve the tone of my writing. The manuscript is now rewritten without changing the message, and I hope that the editor and reviewers (if consulted) will agree that the tone is improved. (Experience indicates that I am a poor judge of this.) I also hope that both the sentence added in response to the comment by Anonymous Referee #2 above, and also the revised terminology (avoiding the term "jets" where possible), will improve understanding.

2. Large body of observations inconsistent with existing model: This is precisely the point of Section 4's final paragraph.  There is indeed a body of observations that the existing model does not explain, namely the decoupling of transpiration and photosynthesis that has been observed widely at very high leaf $T$. The manuscript revised on 26 September cited five articles in this regard, all quite recent; two new citations from 2024 have now been added. Adhering to the diffusion-only model of leaf gas exchanges, plant ecologists have sought purely physiological explanations for such decoupling. I believe that they should take into account how it can result from gas transport processes, and hope that this is one of the clear messages of the manuscript.

3. Quantitative demonstration: Whether or not it was easy to understand, the original manuscript was in fact quite quantitative regarding decoupling. Referring to line numbers in the old revised manuscript (submitted on 26 September), it was explained how:
   a. 121-125: carbon dioxide concentrations are reduced as a consequence of $\Delta e$ (the surplus of water vapour pressure inside substomatal cavities), supressing photosynthesis and
   b. 131-135: for same the gradient in water vapour concentration (or VPD), elevated non-diffusive transport (at high $q$) enhances water vapour egress and so enhances transpiration.

   The newly revised manuscript attempts to make this more explicit to the reader, referring back to suppression of $CO_2$ at the end of the paragraph regarding enhanced transpiration, and just prior to discussing water-use efficiency and decoupling. I considered going into more detail, with additional quantitative examples of decoupling and its magnitude, but feel that this would take away from the concision of the paper.

4. Clearer physical description: Having read (and replied to) the comment by Anonymous Referee #2, Report #1 regarding the Bernoulli principle, supported by these suggestions from Referee X, I realise that I chose my words unartfully (hardly surprising) when publishing the terminology of "stomatal jets" (Kowalski, 2017). The word "jet" has high-velocity connotations and is inappropriate in this context. Air does not jet out of stomata, but rather oozes (which seems a better verb for describing viscous motion). I have therefore revised the manuscript to avoid the use of this unhelpful terminology.

5. What sort of experiments should be done: A new Section 5 ("Prospects for Unveiling Stomatal Fluid Mechanics") has been added prior to the paper's conclusions.

**13 Dec 2024: Co-editor-in-chief decision: Publish subject to minor revisions (review by editor)**

*Thank you for uploading responses to the referees. I feel that in principle the manuscript is publishable but I would appreciate if you note in the text the importance of further studies into the fluid mechanics of the stomatal / atmosphere interface, and to consider the comments by the additional reviewer posted underneath my brief letter from Dec. 12. I think that we all agree that we need to revisit popular assumptions regarding stomatal exchange, and to do so a bit more guidance to the community can make your work be of even greater value to the community.*

As noted above, I have added a new Section 5 suggesting further studies into the fluid mechanics of stomatal gas exchanges, and made several other modifications as consequences of the comments from Reviewer X.